# Novel, Edible Melanin-Protein-Based Bioactive Films for Cheeses: Antimicrobial, Mechanical and Chemical Characteristics

**DOI:** 10.3390/foods12091806

**Published:** 2023-04-26

**Authors:** Ana Rita Ferraz, Manuela Goulão, Christophe E. Santo, Ofélia Anjos, Maria Luísa Serralheiro, Cristina M. B. S. Pintado

**Affiliations:** 1BioISI—Instituto de Biosistemas e Ciências Integrativas, Faculdade de Ciências, Universidade de Lisboa, 1749-016 Lisboa, Portugal; mlserralheiro@ciencias.ulisboa.pt; 2Faculdade de Ciências, Departamento de Química e Bioquímica, Universidade de Lisboa, Campo Grande, 1749-016 Lisboa, Portugal; 3Escola Superior Agrária, Instituto Politécnico de Castelo Branco, 6001-909 Castelo Branco, Portugal; manuela@ipcb.pt (M.G.); ofelia@ipcb.pt (O.A.); cpintado@ipcb.pt (C.M.B.S.P.); 4CATAA—Associação Centro de Apoio Tecnológico Agro-Alimentar, 6000-459 Castelo Branco, Portugal; cespiritosanto@cataa.pt; 5Center for Functional Ecology Science for People & the Planet, TERRA Associated Laboratory, Department of Life Sciences, University of Coimbra Calçada Martim de Freitas, 3000-456 Coimbra, Portugal; 6CEF—Centro de Estudos Florestais, Instituto Superior de Agronomia, Universidade de Lisboa, Tapada da Ajuda, 1349-017 Lisboa, Portugal; 7Centro de Biotecnologia de Plantas da Beira Interior, 6001-909 Castelo Branco, Portugal; 8CERNAS—Centro de Estudos de Recursos Naturais, Ambiente e Sociedade, Instituto Politécnico de Castelo Branco, 6001-909 Castelo Branco, Portugal; 9QRural—Unidade de Investigação Qualidade de Vida no Mundo Rural, Instituto Politécnico de Castelo Branco, Avenida Pedro Álvares Cabral, n° 12, 6000-084 Castelo Branco, Portugal

**Keywords:** cheese, whey protein isolate, bioactive coatings, melanin, edible packing

## Abstract

The cheese rind is the natural food packaging of cheese and is subject to a wide range of external factors that compromise the appearance of the cheese, including color defects caused by spoilage microorganisms. First, eight films based on whey protein isolate (WPI) coatings were studied, of which IS3CA (WPI 5% + sorbitol 3% + citric acid 3%) was selected for presenting better properties. From the IS3CA film, novel films containing melanin M1 (74 µg/mL) and M2 (500 µg/mL) were developed and applied to cheese under proof-of-concept and industrial conditions. After 40 days of maturation, M2 presented the lowest microorganism count for all the microbial parameters analyzed. The cheese with M2 showed the lowest lightness, which indicates that it is the darkest cheese due to the melanin concentration. It was found that the mechanical and colorimetric properties are the ones that contribute the most to the distinction of the M2 film in cheese from the others. Using FTIR-ATR, it was possible to distinguish the rinds of M2 cheeses because they contained the highest concentrations of melanin. Thus, this study shows that the film with M2 showed the best mechanical, chemical and antimicrobial properties for application in cheese.

## 1. Introduction

Cheese undergoes physicochemical, microbiological and biochemical changes during its manufacture, maturation and commercialization. For this reason, packaging plays an essential role in the cheese industry [1,2]. The cheese rind is the natural food packaging of cheese; however, it is subject to a wide range of external factors that can compromise the quality of the cheese, including the appearance of color defects caused by spoilage microorganisms. Several studies have described *Pseudomonas* spp. as an important microorganism with the ability to produce natural pigments in cheese [3,4,5,6,7,8], causing color defects that are undesirable to the consumer and leading to large economic losses for cheese producers since they cannot sell this defective product [5]. 

Consumers are increasingly attentive and concerned about the quality of the food they choose, opting for more natural products without food preservatives, thus causing a rise in interest in packaging concepts that reduce or eliminate the consumption of preservatives [9]. 

Antimicrobial packaging targets surface bacteria suitable for use in packaging cheese and other items where microorganism growth tends to be concentrated on the surface. Active packaging materials that can release active compounds to improve the quality and safety of a wide range of foods during prolonged storage are particularly important [10,11]. Antimicrobial packaging contains only the amount of antimicrobial content necessary to prevent microbial growth, thereby reducing the consumer’s intake of synthetic preservatives that can be substituted with natural preservatives such as organic acids [12]. The cheese-making process has a significant impact on the environment. This process produces large amounts of a by-product called whey, which makes up nearly 90% of the milk used. Whey is normally used to make curd; however, it is also still used as animal feed and fertilizer [13]. 

Proteins, polysaccharides and lipids are the main materials available for producing edible films and coatings. Although the first two are the most widely used and considered the basis of formulations, it is common to find the use of certain plasticizers to reduce brittleness, surface-active agents to aid in film or coating adhesion, and flavors to improve sensorial attributes. In some cases, especially for active food packaging, other compounds are also included, such as antioxidants, antimicrobials, fatty acids and preservatives [14]. 

Whey proteins have attracted increasing interest, not only because they can be obtained from renewable resources (dairy industry by-products), but also due to their recognized oxygen barrier properties [15]. Whey is obtained from cheese and casein manufacturing and contains a great number of proteins with distinctive properties. It is available as a whey protein concentrate (WPC) or whey protein isolate (WPI) according to the protein content, being either 20–80% or >90%, respectively [16]. Both of these options have a good film-forming capacity and present as their main advantage low barrier properties, with oxygen being the most notable (comparable to petroleum-based films) [10]. Several studies have shown that WPI coatings with antimicrobial properties can increase the shelf life of cheeses [10,11]. 

Organic acids are usually used as food additives for preservation due to their antimicrobial activity and acidification capacity [11]. The reason for these antimicrobial activities is related to the chelating activity of organic acids and their ability to dissociate inside the cell and decrease pH. Organic acid activity is also related to pKa, since lower values of pKa lead to a higher decrease in pH. Some works have also showed the synergy between organic acids and other antimicrobials such as natamycin and nisin [11]. 

Information about the mechanical properties of plastics helps to optimize formulations and processes and serves as an object of quality control in their production. Like synthetic plastics, materials composed of biomolecules (bioplastics) can also be analyzed for their mechanical properties using the same methods. The strength, ductility and elastic behavior of the materials are the most important and common mechanical properties to assess when tensile tests are carried out, which can be evaluated by the parameters: maximum tensile stress (TS) and percentage of elongation at the rupture (E%). TS corresponds to the maximum stress value applied to the material during the test, and E% corresponds to the value of the deformation, as a percentage, when the sample is ruptured [17].

Pigments are compounds of importance in several industries, such as the food industry, where they can be used as additives, color intensifiers and antioxidants [5,18]. As the current trend around the world has shifted to the use of eco-friendly commodities, the demand for natural dyes is increasing. 

Melanins are a unique class of natural pigments that can be considered functional biocompounds for multiple potential industrial applications, such as in materials science, biomedicine and cosmetics [5]. Melanins tend to be either black or brown pigments, although other colors may occur [19]. Melanins derived from L-3,4-dihydroxyphenylalanine (L-DOPA), a tyrosine-derive amino acid, are referred to as eumelanins and are black or brown [19].

Melanins are pigments that are produced by several microorganisms, such as molds, yeasts and bacteria. *Pseudomonas putida* ESACB 191, which was isolated from goat cheese rind, was described as a brown eumelanin producer by Ferraz et al. (2021) [5]. The bioactivity of melanin was evaluated as the capacity for scavenging free radicals (antioxidant activity) with an EC_50_ of 74.0 ± 0.2 μg/mL, and as an acetylcholinesterase inhibitor with an IC_50_ of 575 ± 4 μg/mL [5]. 

This study aimed to develop a melanin WPI coating formulation with antimicrobial activity capable of acting on a wide spectrum of microorganisms, thus contributing to the increase in the shelf life of cheese. 

## 2. Materials and Methods

### 2.1. Preparation of WPI Coatings and Films

A screening of the filmogenic solution was made that presented the best mechanical, color and antibacterial properties. For this, eight formulations were studied based on the whey protein isolate (WPI) with two different plasticizers (sorbitol and glycerol) and two acidity regulators, citric and lactic acid. The scheme of the preparation of the eight coatings is shown in Table 1.

The methodology applied was generally described by Pintado et al. (2009) [11], with some modifications. In total, 5 g of WPI (92% minimum protein content, dry basis; Carbery Group, Ballineen, Ireland) was dissolved and homogenized in 100 mL of distilled water and stirred with a magnetic bar for 15 min. For the plasticizer, 3.0 g of glycerol (VWR International, Radnor, PA, USA) or sorbitol (Merck, Darmstadt, Germany) was introduced to the solution and stirred again for 15 min. Each organic acid has an acidity regulator and antibacterial functions; thus, lactic acid (Merck, Germany) and citric acid (VWR International, USA), at 3.0% (*w*/*v*), were added, and the final solution was stirred for 15 min until uniform solutions were obtained. After measuring the pH, the solutions were heated at 90 °C for 30 min in a shaking water bath. 

After selecting the WPI coating with the best characteristics, this coating was used as a basic formulation to which nisin was added as the antilisterial, natamycin as the antifungal and melanin as the antimicrobial, antioxidant, and dye, which will be described later. Once cooled to room temperature, an adequate volume of nisin solution at 10^5^ IU/mL (Sigma-Aldrich Chemical Co., St. Louis, MO, USA), previously made with 20 mM HCl, was incorporated to produce a final concentration of 50 IU/mL. 

This concentration was previously determined as the lowest concentration tested that completely inhibited the growth of *Listeria monocytogenes* strains used in this assay (data not shown), which was described by Pintado et al. (2009). The commercial, normally used natamycin (Sigma-Aldrich Chemical Co., St. Louis, MO, USA) was incorporated in a concentration of 3.5 mg/mL. The melanin used was produced by *Pseudomonas putida* ESACB 191, according to the research of Ferraz et al. (2021) [5]. The melanin was used as a colorant and antioxidant agent and was used in concentrations of 74.2 µg/mL (EC_50_ for DPPH scavenging activity) and 500 µg/mL of pure melanin [5]. The films were produced by weighing 10 g of the filmogenic solution into 90 mm polystyrene disposable plates. After drying for 24 to 48 at 37 °C, the films were peeled from the plates and stored at room temperature until they were used. 

### 2.2. Screening of WPI Films’ Antimicrobial Activity 

The antimicrobial activity of films was tested in 17 strains (9 bacteria strains, 5 molds and 3 yeasts), of which 12 belonged to the microbial culture collection of the Laboratory of Microbiology at the Agrarian School of the Polytechnic Institute of Castelo Branco, Portugal. The remaining five were *Pseudomonas aeruginosa* ATCC 27853, *Pseudomonas fluorescens* ATCC 13525, *Staphylococcus aureus* ATCC 25923, *Listeria monocytogenes* NCTC 11994 and *Candida albicans* ATCC 10231. The wild strains came from the dairy factory and their origin is presented in Section 3.2. Antimicrobial susceptibility testing was performed according to the EUCAST disk diffusion method [20] (EUCAST 2020), with some modifications.

#### 2.2.1. Antibacterial Test

The films were aseptically cut into discs (6 mm in diameter) and placed on the surface of the plates with the appropriate growth medium for each strain. For *Pseudomonas* spp., *Serratia* spp. and *Staphylococcus aureus* ATCC 25923, Nutrient Agar (Oxoid, Hants, United Kingdom) was used. Tryptone Soya Yeast Extract Agar (Oxoid, Hants, UK) was used for *Listeria monocytogenes* NCTC 11944. 

The plates were pre-inoculated with a 0.5 McFarland-scale cell suspension for bacteria strains, and a 1.0 MacFarland scale for yeasts. After 24 h of incubation for bacteria and 48 h for yeasts at the respective growth temperatures, the zones of inhibition that formed around the film disks were measured with a digital micrometer (0–200 mm digital caliper). The assay was performed in triplicate.

#### 2.2.2. Antifungal Test

Fungal strains were grown in Potato Dextrose Agar (Sigma-Aldrich Chemical Co., St. Louis, MO, USA) for 7 days at 25 °C. Mold was removed from the Petri dish with the aid of a swab and placed into a vial with 30 mL of physiological saline solution (sodium chloride 0.85%). The spore suspension was filtered through a funnel with about two layers of sterilized gauze to remove the mold mycelium. The optical density of the filtered suspension was measured in the spectrophotometer at a wavelength of 570 nm and set to values between 0.15 and 0.17. 

The films were aseptically cut into discs (6 mm in diameter) and placed on the surface of the plates. After 3–4 days of incubation at 25 °C, the zones of inhibition were measured. The test was carried out in triplicate.

### 2.3. Physical Property Analysis

#### 2.3.1. Mechanical Properties

The tensile properties of eight WPI films were evaluated according to ASTM D882-02: “Standard test method for tensile properties of thin plastic sheeting” [21]. Strips were cut to be 1 cm wide and 6 cm long. These strips were then placed for at least two days in a desiccator with a saturated solution of magnesium nitrate hexahydrate (Applichem, Darmstadt, Germany) in the background, allowing us to create an atmosphere with an RH of 50 ± 5% and temperature of 25 ± 2 °C. The thickness of the strips was measured in three different positions using a digital micrometer (0–200 mm digital caliper). In total, 10 strips (samples) were analyzed for each kind of film.

The TA.XT plus texture machine (Stable Micro Systems, Godalming, UK) equipped with traction grippers (model A/TG) was used to evaluate the mechanical properties. The strips were mounted on the grippers with an initial separation between them of 3 cm, and the test was performed at a speed of 5.0 mm/s until the strips broke. Throughout the test, the Texture Exponent 32 software connected to the test machine built a graph of the pull force applied by the machine versus the gripper separation distance (in mm). 

The strength, ductility and elastic behavior of the materials are the most important and common mechanical properties to evaluate when performing tensile tests [22]. 

Tension is a pressure and, as such, corresponds to the application of a force per unit of area. Knowing the thickness of the strips and their width, we calculated the cross-sectional area; as this cross-sectional area was the surface on which the tensile force was applied, calculating the *TS* was enough for dividing the maximum force by this area. Equation (1) represents the formula for calculating *TS* (in MPa), where *F* is the maximum tensile force (in N), h is the strip thickness (in mm) and L is the strip width (in mm): (1)$$TS=\frac{F}{h\times L}$$

*E*% is calculated in relation to the initial length of the sample. According to Equation (2), the variables *Cf* and *Ci* are the distance between the grippers at the break and the initial length of the strip, respectively.
(2)$$E\%=\frac{Cf-Ci}{Ci}\times 100$$

For each strip, a graph was obtained from which the necessary values were obtained for calculating the parameters of tensile strength (*TS*) and the percentage of elongation at the break (*E*%).

#### 2.3.2. Color Analysis Properties 

The color analysis was performed with the CIELAB color space. The CIELAB color space, also referred to as L*a*b*, is a color space defined by the International Commission on Illumination (CIE) in 1976 [23,24]. 

The CIELAB color space is the most frequently used system to specify food colors [24]. It is a three-dimensional Cartesian space with three mutually perpendicular color coordinates: L*, the correlate of perceptual lightness; a*, which represents the red (a* > 0)–green (a* < 0) axis; and b*, which represents the yellow (b* > 0)–blue (b* < 0) axis [23]. The film’s color was measured using a colorimeter (CR-321, Konica Minolta, Tokyo, Japan), and five measurements for each type of film were made. As the standard, an empty base of a clean Petri dish was used. For coated cheese color measures, the standard that was used was the cheese without treatment. The results (mean ± standard deviation) were expressed as L*, a* and b*. Additionally, ∆E (color difference) was also calculated using Equation (3), as follows:(3)$$∆E*=\sqrt{{\left(L_{\mathrm{standard}}-{\;L}_{\mathrm{sample}}\right)}^{2}+{\left(a^{*}{}_{\mathrm{standard}}-{\;a}^{*}{}_{\mathrm{sample}}\right)}^{2}+{\left(b^{*}{}_{\mathrm{standard}}-{\;b}^{*}{}_{\mathrm{sample}}\right)}^{2}}$$

### 2.4. Application of Melanin WPI Coatings under Industrial Conditions

#### 2.4.1. Melanin WPI Cheese Coatings

The WPI coatings presented in Table 2 were applied to goat cheese made with raw milk after 48 h of maturation. The coatings were applied using the dipping technique that is already used in the cheese industry. For this assay, cheeses were coated with the novel melanin films M1 and M2, and these were compared with the commercial film (CF) used in the cheese industry, which is composed of RIOCOBERT (biopolymer E-466) and VIPLAST-1AX (vinyl polymer) at the concentrations indicated by the manufacturer, as shown in Table 2. For the evaluation of the antimicrobial activity of the films, the commercial films were used separately. For the application of the commercial films on the cheese, the methodology used in the cheese factory was applied. The RIOCOBERT was applied at 48h as the other coatings under study and VIPLAST-1AX was applied to the cheeses already coated with RIOCOBERT after 20 days of maturing, at the transition from chamber 1 to chamber 2. The assay was carried out in triplicate, and the monitoring of the cheese’s maturation was carried out for 40 days. 

The methodology design of the proof of concept in a cheese factory is presented in Figure 1. 

After coating, the cheeses were placed for 20 days in a maturation chamber (C1) at 6 ± 1 °C and 99% RH. After this maturation period, the cheeses were moved to a second chamber (C2), which was a drying chamber at a temperature of 12 ± 1 °C and 80% HR, until the end of maturation (40 days). 

After 40 days of the cheese maturation period (T40), the rinds of the cheeses where the applied films were located were subjected to various analyses (antimicrobial, color and FTIR).

#### 2.4.2. Melanin WPI Antimicrobial Activity

The antimicrobial activity of the films was evaluated after the 40-day cheese maturation period (T40). The antimicrobial activity of the films was compared with that of the T0 cheeses, which corresponds to cheese within 48 h of maturation without any treatment. To understand the film antimicrobial activity, the times T0 and T40 were the periods used. The microorganism enumeration at 30 °C (CFU/g) was performed according to ISO 4833-1:2013 [25]; the coagulase-positive staphylococci enumeration (CFU/g) was carried out according to NP 4400-1:2022 [26]; *Listeria monocytogenes* detection at 25 g was carried out according to ISO 11290-1:1996 [27]; *Pseudomonas* spp. enumeration (CFU/g) was carried out according to ISO 11059:2009 [28]; and the mold and yeast enumeration (CFU/g) was carried out according to NP 3277-1: 1987 [29]. For the detection of *Listeria monocytogenes*, 25 g of cheese rind (1 cm thick) was weighed and placed into a Stomacher bag, where 225 mL of the first enrichment broth was later added. For counting the different groups of microorganisms, 10 g of cheese rind (1 cm thick) was weighed into a different Stomacher bag and 90 mL of diluent was added. The homogenization of the samples was carried out in the Stomacher 400 Circulator (Seward) for 90 s at 260 rpm. Each sample was analyzed in triplicate for each parameter. 

### 2.5. FTIR Analysis

Approximately 10 g of each representative cheese sample was taken and very well mixed. The cheese spectra were acquired via an FTIR-ATR spectrometer (ALPHA, Bruker Optik GmbH, Ettlingen, Germany) with 32 scans per sample and a resolution of 4 cm^−1^ in the wavelength region of 4000–400 cm^−1^, using diamond single reflection. The experiments were carried out at room temperature. The background measurement was made using air.

The FTIR-ATR systems were operated using the OPUS 7.5.18 BRUKER software provided by the manufacturer.

### 2.6. Statistical Analysis

The statistical differences between groups were calculated with the one-way analysis of variance (ANOVA), and STATISTICA 7 (StatSoft Inc. United States, Tulsa, OK, USA) was used. For spectral data analysis and principal component analysis (PCA), OPUS^®^ version 7.5.18 (Bruker Optics, Ettlingen, Germany) and UnscramblerX 10.5 (CAMO, Oslo, Norway) were used.

Prior to PCA analysis, some spectral pre-processes were tested, namely, multiplicative scatter correction (MSC); minimum maximum normalization (MinMax); vector normalization (VecNor); straight line subtraction (SLS); first derivative (1stDer); second derivative (2ndDer); and some combinations of these pre-processes.

## 3. Results and Discussion

### 3.1. Screening of Mechanical Properties of WPI Films

#### 3.1.1. Tensile Strength, Thickness and Extensibility of WPI Films

Cheese producers have shown a need for a completely edible cheese which has no dark spots on the surface. Therefore, this study aimed to develop a new cheese coating based on a WPI that would support various bioactivities in order to avoid unwanted staining of the cheese and would be a colored film that meets the consumer’s color requirements. 

The WPI concentration used was the result of a screening where the 5% concentration was the one that showed better mechanical and visual properties The concentration of the organic acids (3%) was selected after considering the results of the previous study by Pintado et al. (2009) [11].

Thus, eight coating formulations with different concentrations (3 and 5%) of glycerol and sorbitol plasticizers combined with citric and lactic acids were tested for the first time.

Figure 2 shows the mechanical properties tested, including tensile strength (*TS*), thickness and elongation at the break (*E*%) of eight WPI films. 

Tensile strength is the maximum pull that can be achieved before the film breaks/tears. Tensile strength properties depend on the concentration and type of edible film composition [30]. The analysis results of the variance showed that the treatment of different glycerol and sorbitol concentrations in the WPI edible film formulations significantly affected (*p* ≤ 0.05) the tensile strength. The average values of edible film tensile strength can be seen in Figure 2.

Film thickness is an important characteristic in determining the suitability of edible films as food product packaging. Thickness can affect the mechanical properties of edible films, such as tensile strength and elongation [31]. 

The thickness of an edible film is an important parameter that influences the use of edible film in product packaging [32]. The thickness values of WPI films with glycerol as a plasticizer were lower than those of WPI films with sorbitol. The value of IS3CA was higher than those of the other films. The thickness of the films was significantly influenced (*p* ≤ 0.05) by the plasticizer that was used. In this case, sorbitol (3%) increased the film thickness. 

Films that contained glycerol were less extensible than films that contained sorbitol [33], as shown in Figure 2. The concentration of 3% of plasticizers and organic acids showed better mechanical properties than a concentration of 5%. 

Tensile strength was affected by the plasticizer and organic acid. Figure 2 shows that sorbitol as a plasticizer increased the TS compared to glycerol films at a confidence level of 95%. Films were prepared using equal mass ratios of sorbitol or glycerol to the whey protein isolate. Thus, films plasticized with glycerol contained more plasticizer for the mole basis than the films plasticized with sorbitol because the molecular weight of sorbitol (182.14 g/mol) was almost exactly twice that of glycerol (92.09 g/mol). 

The higher the sorbitol concentration used, the lower the tensile strength was. This was due to the fact that sorbitol as a plasticizer can reduce the energy required for the molecule to move, causing its stiffness and tensile strength to decrease [34]. In addition, sorbitol can also reduce internal hydrogen bonds and increase the molecular distance, causing the structure of the film formed to become softer and more flexible [31,35]. The highest tensile strength of the edible film (4.78 MPa) was obtained with IS3CA. 

Edible films that have high tensile strength will be able to protect the packaged products well from mechanical disturbance [36].

The thickness values of WPI films with glycerol as a plasticizer were lower than those of the WPI films with sorbitol. The value of IS3CA was higher than those of the other films. The thickness of the films was significantly influenced (*p* ≤ 0.05) by the plasticizer that was used. In this case, sorbitol increased the film thickness. However, the organic acids (citric and lactic acids) did not influence the film thickness. WPI films with sorbitol and citric acid were the films with higher thicknesses. 

The elongation percentage is a length change percentage of the edible film when it is pulled to break [35]. The greater the elongation percentage value, the better the edible film is because it is more elasticity and is not easily torn [37].

The organic acids were expected to have some influence on the mechanical properties of the films. The elongation presented values that ranged between 13.18 and 18.15% for glycerol-plasticized films and between 7.95 and 18.18% for sorbitol-plasticized films (Figure 2). 

However, the presence of organic acids also influenced the elongation of the WPI films. CA films were more extensible than LA films with both glycerol and sorbitol, exhibiting a confidence level of 95%. However, citric acid increased the elongation of glycerol and sorbitol WPI films. Organic acids could have a plasticizing effect because they are small molecules with hydroxyl groups [38]. Citric and lactic acids have four and two hydroxyl groups, respectively. The hydroxyl groups of the plasticizer replaced the polymer–polymer interactions by developing polymer–plasticizer hydrogen bonds, which probably explains the higher elongation values for the films produced with citric acid. Electrostatic interactions around the pI are responsible for protein aggregation, and consequently, proteins are less flexible and therefore less likely to form a cohesive film [11].

As the IS3CA film showed the best mechanical properties, it was selected as a base formulation for two novel melanin WPI films. Pure melanin produced by *Pseudomonas putida* ESACB 191 was added, as it showed antioxidant and coloring activity as described by Ferraz et al. (2021) [5]. Melanin was used at concentrations of 74 µg/mL, as it showed antioxidant activity, and 500 µg/mL, as it inhibited the enzyme acetylcholinesterase, as was described by the authors Ferraz et al. (2021) in a previous work that also showed that melanin was a eumelanin with a brown color [5].

To increase the bioactivity of the novel melanin WPI films, the natural preservatives nisin and natamycin were added [39]. The concentrations of food additives were used according to European law. The nisin concentration was described by Pintado et al. (2009) [11], and the natamycin concentration was the same as that used in commercial coatings for cheese.

Before applying the novel melanin WPI films to cheese, the mechanical properties of the novel M1 (WPI 5% + sorbitol 3% + citric acid 3% + natamycin (3.5 mg/mL) + nisin (50 IU/mL) + melanin (74 µg/mL)) and M2 (WPI 5% + sorbitol 3% + citric acid 3% + natamycin (3.5 mg/mL) + nisin (50 IU/mL) + melanin (500 µg/mL)) films were studied and compared with a commercial film (CF). This commercial film was composed of RIOCOBERT and VIPLAST-1AX. Figure 3 shows the mechanical properties tested: tensile strength (*TS*), thickness and elongation at the break (*E*%) of the novel melanin WPI films. 

The analysis of variance showed that there were no significant differences between the mechanical properties of M1 and M2 (*p* ≤ 0.05). However, the novel melanin WPI films M1 and M2 showed significant differences when compared with the CF at a 95% confidence level. This result was already expected, since the chemical compositions of the novel M1 and M2 films are similar to each other, but differ from the chemical composition of the CF. However, the CF, M1 and M2 films did not show significant differences in thickness (*p* ≤ 0.05), which indicates that the new melanin WPI M1 and M2 films have a similar thickness to the CF film already used by the cheese industry. 

#### 3.1.2. Color

Color attributes are of prime importance because they directly influence product appeal and consumer acceptability.

CIEL*a*b* color values, i.e., the total color difference (ΔE) of the WPI films from glycerol and sorbitol as plasticizers, as well as from the organic acids of citric and lactic acids, are shown in Figure 4. 

IS3CA was the film that showed a higher L* value than the others at a confidence level of 95%. However, this film also showed lower a*, b* and ΔE values. 

All of the films were transparent, and thus it was normal for them to have higher L* values. Plasticizers such as glycerol and sorbitol, which are known as polyols, are commonly used additives and are most popular partly due to their low cost but are primarily used because they provide films with flexible characteristics by reducing the hydrogen bonds in the polymer chains and increasing the molecular space which, in turn, adds a better visual appearance [40,41].

### 3.2. Antimicrobial Activity of Melanin WPI Films

The results of the influence of the antibacterial activity due to the addition of citric and lactic (3%) acids, the plasticizers glycerol and sorbitol (3%), natamycin (3.5 mg/mL), melanin M1 (74 µg/mL) and melanin M2 (500 µg/mL) to the filmogenic solution are shown in Table 3.

Table 3 shows that there were significant differences between the antimicrobial activity of glycerol and sorbitol films at a confidence level of 95%. No significant differences in antimicrobial activity were observed for the IG3CA and IG3LA films. 

However, the IS3CA films had significantly higher antimicrobial activity than the IS3LA films at a confidence level of 95%. Organic acids with a low molecular weight had higher antimicrobial activity. The organic acids had an antibacterial effect due to the ability of undissociated molecules to pass through the cell membrane and ionize to release protons in the cytoplasm, thereby depressing the intracellular pH and inhibiting the metabolism [42].

The IS3CA film stood out from the other films as it showed a higher antimicrobial activity at a confidence level of 95%. This result was because the pKa of citric acid is lower than the pKa of lactic acid, and consequently, at similar concentrations, citric acid tends to decrease the pH more than lactic acid [11]. The greater inhibitory effect also likely reflects the fact that the reduced ability of citric acid to enter bacterial cells is compensated for by its enhanced ability to dissociate inside the cell, and thus decrease the internal pH [11]. Furthermore, a study showed that in WPI biodegradable films, citric acid was added to increase the antimicrobial activity and plasticizing to improve the mechanical properties [27]. 

To prevent fungal growth in cheese rinds, natamycin (3.5 mg/mL) was incorporated as an antifungal agent, as it is already used in the cheese industry; then, the antimicrobial activity of these new films was tested. The incorporation of natamycin into the formulation statistically increased (*p* ≤ 0.05) the antifungal activity of the film (Table 2). Only the yeasts *Candida albicans* ATCC 1231 and *Candida zeylanoides* were not susceptible to the IS3CAN film. 

In addition, this IS3CA formulation (WPI 5% + sorbitol 3% + citric acid 3%) was selected as the basic formulation for developing novel films with melanin. It was found that the films IS3CA, IS3CAN and IS3CAM were statistically different from each other (*p* ≤ 0.05). However, the addition of melanin increased the antimicrobial activity of the film for an average of 78% of bacterial strains. 

Melanin produced by *Pseudomonas putida* ESACB 191 was added to the IS3CA film to increase its bioactivity [5]. The M1 (74 µg/mL) and M2 (500 µg/mL) melanin films were statistically different and differed from the IS3CA, IS3CAN and IS3CAM films at a 95% confidence level. These new melanin films of M1 and M2 are intended to guarantee public health and control the presence of *Listeria monocytogenes* at legal values. To this end, nisin was added as an antilisterial agent to the new melanin films, according to Pintado et al. (2009) [11]. The incorporation of nisin and natamycin can justify the statistical difference between the films, as well as the concentration of melanin. It was found that M2 was the film that showed the highest antimicrobial activity, even when compared to the RIOCOBERT and VIPLAST-1AX commercial films. Thus, M2 was found to be a new, very promising melanin edible film that can be applied to cheeses.

### 3.3. Novel Melanin Coatings in Cheese Maturation 

After selecting the base formulation of the WPI film (IS3AC), two novel melanin coatings (M1 and M2) containing melanin (dye and antimicrobial), natamycin (antifungal) and nisin (antilisterial) were developed. For the proof of concept, these melanin films were compared to the commercial films of RIOCOBERT and VIPLAST-1AX, which are nonedible coatings that also contain natamycin in their composition. 

The melanin coatings prepared based on IS3CA were applied to the cheeses as a proof of concept under industrial conditions. The experiment was carried out in triplicate, and it was intended to evaluate the differences in antimicrobial activity, color and chemical characteristics present on the surfaces of cheeses treated with different coatings (CF, M1 and M2) and on the surfaces of cheeses without any treatment (C). Figure 5 shows the visual evolution of cheese maturation over the 40 days of ripening.

This proof of concept was carried out to apply these two formulations of melanin WPI films (M1 and M2) with the intent of comparing them with each other, in addition to comparing them with the commercial formulation (CF) used in the industry and with the cheese without any coating (C) that functioned as the control of the assay. After maturation (40 days), the cheeses were subjected to several laboratory tests, namely, microbiological analyses to evaluate the antimicrobial capacity of the films tested and the color of the rinds of the coated cheeses to understand the color variation in the cheese rind conferred by the different films applied to the cheese, as well as the characterization of the cheese rind via FTIR-ATR. 

#### 3.3.1. Film Antimicrobial Activity in Cheese

To perceive the evolution of the microbial load in the cheese rind, analyses were conducted on the cheese before the application of the tested coatings (T0) and after 40 days (T40) of maturation of the cheeses with the tested coatings. The analytic results are of microbial growth on the rind of the cheeses during maturation are shown in Figure 6.

Figure 6 shows that the microorganisms at 30 °C increased in all treated cheeses after 40 days of maturation when compared to T0. For the remaining microbiological parameters analyzed, it was verified that the M1-coated and M2-coated cheeses showed an average reduction of 16% and 21% of Log CFU/g, respectively, when compared with CF-40. There was a higher reduction of 49% of Log CFU/g of *Pseudomonas* spp. for M2-40 compared with CF-40. However, of the new melanin edible films (M1 and M2), M2 presented the lowest microorganism count for all parameters analyzed, as shown in Table 3. Thus, the M2 melanin coating proved to be a promising antimicrobial edible coating for application in cheese packaging.

#### 3.3.2. Color

Color is one of the most important attributes that will be first seen by consumers before eating the cheese, causing it to be a primary consideration of consumers when making purchasing decisions [23]. The color is also an indicator of the cheese’s quality, along with attributes such as flavor and ripeness [43]. 

CIEL*a*b* color values, the total color difference (ΔE) of the cheese rind of the control (C) that is uncoated, the cheese coated with the commercial formulation (CF), the cheese coated with melanin film 1 (M1) and the cheese coated with melanin film 2 (M2) after 40 days of ripening, are shown in Figure 7. The ΔE provides a good measure of the color difference since it considers all three color parameters: lightness (L*), the red–green (a*) axis and the yellow–blue (b*) axis [44,45,46]. 

According to the ΔE values, there are significative differences between the coated cheeses at a confidence level of 95%. The addition of melanin caused a decrease in the L* parameter (lightness) (*p* ≤ 0.05). M2 became darker, which can be attributed to the presence of a higher concentration of melanin (500 µg/mL). All cheeses with and/or without films had negative values of the parameter a*. The negative values of the parameter a* represent the presence of green, whereas the positive values of a* are attributed to red [23,24]. All tested cheeses exhibited positive values for parameter b*, which indicates the presence of yellow. For all coated cheeses, a* and b* values were lower than L*. Thus, lightness was the main color component in all films. In Figure 7, it is possible to verify that the highest ΔE is that of M2, which stands out due to its color difference and which may be related to lightness (L*) since it was the cheese that presented the lowest L*. This indicates that it was the darkest cheese due to its melanin concentration (500 µg/mL). 

Visually, the films with the addition of melanin had a light-brownish color that was similar to that of traditional matured cheeses, meeting the consumers’ expectations.

Figure 8 shows the PCA plot (mean-centered) of the ten variables (microbial, color and mechanical properties) obtained with C, CF, M1 and M2. This analysis showed that three components explained 84.2% of the total variation.

Figure 8a shows that PC1 (33% of the total variation) distinguishes C from the other modalities and that PC2 (27.3% of the total variation) separates the properties of M1 and M2. It was found that the mechanical (TS, T and E%) and colorimetric properties (a* and b*) were the ones that most contributed to the variation between CF and M2. PC3 separated CF from M2 (23.6% of total variation), based on TS, a*, E% and b*, as shown in Figure 8b. M2 stood out from the rest for its TS, since it had better mechanical and color properties than the rest.

M2 contained in its formulation WPI, sorbitol, citric acid, melanin, nisin and natamycin, which makes this food coating 100% edible and means that it can be used to coat cheese, thus avoiding food waste. Although it contains antifungal natamycin, it is also present in CF, which is a plastic coating (vinyl polymer) that is not edible and can be toxic when consumed, making the use of this type of coating dangerous for human and environmental health [47,48].

### 3.4. FTIR of Melanin WPI Cheese Coating

Figure 9 shows the FTIR-ATR spectra of cheese rind samples, representing the spectral bands arising from the characteristic functional group vibrations of the matrix. 

It was observed in the analyzed samples that peaks were present in the regions of 3500–2800 and 1750–700 cm^−1^. Between 3325 and 3081 cm^−1,^ it was possible to observe an alteration in the peak intensity related to the elongation of hydroxyl groups (OH stretching), while the predominant peaks located at 2921 and 2853 cm^−1^ were attributed to the asymmetrical and symmetrical stretching vibrations of the fatty acid groups (CH_2_ and CH_3_). It was also possible to observe a continuous increase in several peaks (1743 cm^−1^, stretching symmetrical (C=O); 1461 cm^−1^, angular deformation (CH_3_); 1377 cm^−1^, angular deformation (CH_2_); and 1236, 1156, 1112 and 1099 cm^−1^, stretching asymmetrical/symmetrical (C–O)) of acids, esters and aliphatic chains of fatty acids and carbohydrates [49,50,51].

According to [51,52], most peaks identified in the middle infrared region during cheese analysis can be identified and attributed to functional groups such as CH_2_/CH_3_, C–O and C=O related to triacylglycerides and ester bonds. The peaks between 1700 and 1496 cm^−1^ indicate two important regions for the analyzed fractions: amide I (1631 and 1629 cm^−1^), which is associated with the symmetrical stretching of the C=O and C−N functional groups, and amide II (1575 and 1537 cm^−1^), which is attributed to the vibration of the N−H angular deformation and C–N stretching functional groups. 

Amide-related peaks describe the behavior of secondary protein structures, which, during maturation, lose polypeptide chains that release peptides and amino acids, resulting in alterations to the α-helix, β-leaf and β-curve structures that explain these peaks in the control (C) spectra. The spectrum of the commercial formulation (CF) did not show peaks at 1631 and 1537 cm^−1^ since its chemical nature is based on carboxymethylcellulose. On the other hand, the cheese rinds with M1 and M2 are based on a whey protein isolate (WPI) which contains >90% protein. However, the spectrum of M1 did not contain peaks at 1621 and 1537 cm^−1^, as was evident in the M2 spectrum. This is justified by the difference in the melanin concentration present. According to Ferraz et al. (2021), the melanin spectrum also contains these two peaks [5]. As M2 has 6× more melanin, it was expected that the peaks in M2 would be more evident than in M1. 

Figure 10 shows the PCA plot (mean-centered) of the FTIR-ATR spectra obtained with C, CF, M1 and M2. This multivariate statistical method was used to discriminate the differences between samples in terms of qualitative analysis and reveal that this technique can discriminate between the differences in the chemical structure of the cheese rinds. 

The PCA was performed in the spectral region from 3200 cm^−1^ to 500 cm^−1^ after testing for several pre-processes.

The first two components justified 95% of the total variance (Figure 10a). The first component explained 64% of the variance and was capable of distinguishing between M1, M2 and CF, which is justified by the chemical nature of the three films. The CF was made with plastic (vinyl polymer) and carboxymethylcellulose (E-466), while M1 and M2 were formulated based on a WPI. However, distinguishing M1 from M2 shows that the concentration of melanin present in the film chemically influences the structure of the cheese rind. The second component explains 31% of the variance and distinguishes the control from the other samples.

According to the loading graphic of PC-1 (Figure 10b) of cheeses that was plotted, the sample separations in the regions from 3200 cm^−1^ to 2700 cm^−1^ and from 1800 cm^−1^ to 1000 cm^−1^ were the more discriminating ones.

The most important region was between 1700 and 1496, as it was related to the C=O and C–N groups of the melanin present in the rinds of both M1- and M2-coated cheeses. This region distinguished the rinds of M2 cheeses because they were the ones that contained the most melanin. 

Thus, it was verified that melanin is an important element of distinction allowing for the control of the quality of cheeses.

The PCA performed with the spectral information showed similar results to those made with the analytical data, with only some differences because of the matrix effect. By performing FTIR-ATR, we obtained more analytical information.

## 4. Conclusions

In this study, new bioactive WPI films with melanin were developed to coat cheeses. Eight different films were tested, and IS3AC was selected as the basis for the development of films with melanin acting as an antioxidant and dye, nisin acting as an antilisterial and natamycin acting as an antifungal. 

The new melanin films, M1 and M2, were applied to cheeses in a pilot test under industrial conditions and compared with a commercial film (CF) (RIOCOBERT and VIPLAST-1AX) used in the industry and with a control (C) cheese without any treatment. After 40 days of maturation, the M2-treated cheese presented the lowest microorganism count for all the microbial parameters analyzed. The highest ΔE was found for M2, causing it to stand out because of its color difference, which may be related to its luminosity (L*) since it was the cheese that presented the lowest L*. This indicates that it was the darkest cheese due to the melanin concentration (500 µg/mL) present in M2. It was found that the mechanical (TS, T and E%) and colorimetric properties (a* and b*) were the ones that contributed the most to the distinction of the M2 film on cheese. FTIR-ATR showed that the most important region was between 1700 and 1496, which was related to the C=O and C–N groups of melanin present in both the rinds of M1- and M2-coated cheeses and allowed us to distinguish the rinds of M2 cheeses because they were the ones that contained the most melanin. Thus, it was verified that melanin is an important element of distinction that enables us to control the quality of cheese. Thus, this study showed that the M2 film with melanin was the film that showed better mechanical, chemical and antimicrobial properties for application in the cheese industry. The use of this 100% edible film is a quick and sustainable solution for controlling the quality and food safety of cheeses. This study presents a new active packaging option for cheeses that is completely edible and has an appealing color for the consumer.

## Figures and Tables

**Figure 1 foods-12-01806-f001:**
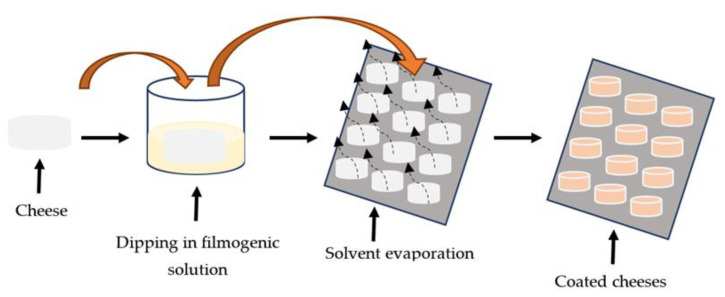
Assay design of application of melanin WPI coatings in cheese under industrial conditions.

**Figure 2 foods-12-01806-f002:**
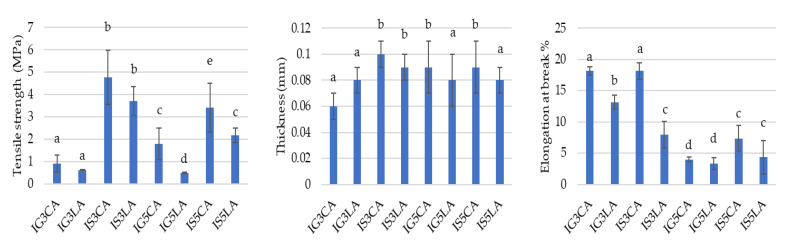
Mechanical properties of WPI-based films: IG3CA—WPI 5% + glycerol 3% + citric acid 3%; IG3LA—WPI 5% + glycerol 3% + lactic acid 3%; IS3CA—WPI 5% + sorbitol 3% + citric acid 3%; IS3LA—WPI 5% + sorbitol 3% + lactic acid 3%; IG5CA—WPI 5% + glycerol 5% + citric acid; IG5LA—WPI 5% + glycerol 5% + lactic acid; IS5CA—WPI 5% + sorbitol 5% + citric acid 3%; IS5LA—WPI 5% + sorbitol 5% + lactic acid 3%. Different lowercase letters (a–e) represent significant differences (*p* ≤ 0.05).

**Figure 3 foods-12-01806-f003:**
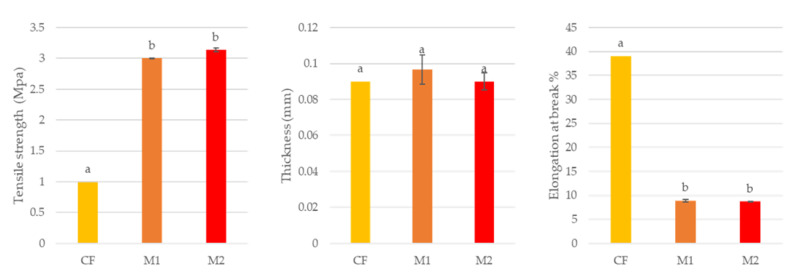
Mechanical properties of films for coated cheese: CF—commercial film (RIOCOBERT and VIPLAST-1AX); M1—WPI 5% + sorbitol 3% + citric acid 3% + natamycin (3.5 mg/mL) + nisin (50 IU/mL) + melanin (74 µg/mL); M2—WPI 5% + sorbitol 3% + citric acid 3% + natamycin (3.5 mg/mL) + nisin (50 IU/mL) + melanin (500 µg/mL). Different lowercase letters (a–b) represent significant differences (*p* ≤ 0.05).

**Figure 4 foods-12-01806-f004:**
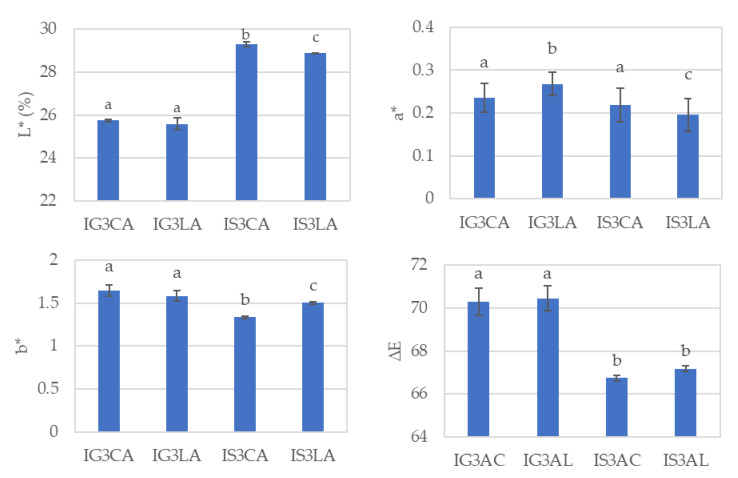
Color parameters and total color differences of WPI films from film-forming solutions subjected to different formulations: IG3CA—WPI 5% + glycerol 3% + citric acid 3%; IG3LA—WPI 5% + glycerol 3% + lactic acid 3%; IS3CA—WPI 5% + sorbitol 3% + citric acid 3%; IS3LA—WPI 5% + sorbitol 3% + lactic acid 3%. Different lowercase letters (a,b,c) represent significant differences (*p* ≤ 0.05).

**Figure 5 foods-12-01806-f005:**
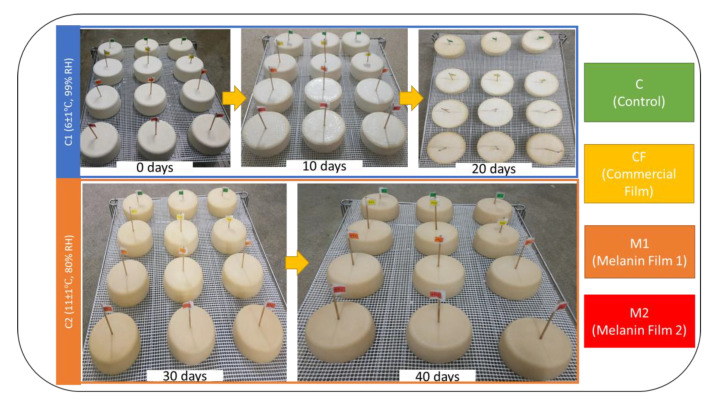
Proof-of-concept cheese evolution with different coatings during 40 days of ripening. The little flags sticking out of the cheese rinds indicate the type of coating tested.

**Figure 6 foods-12-01806-f006:**
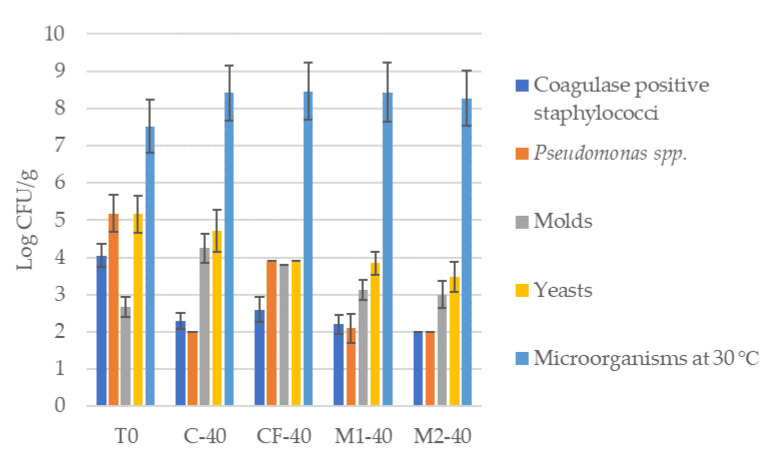
Microbial analysis of cheese rinds with coatings. T0—cheese without coating at 0 days of maturation. C-40—cheese control after 40 days of maturation (cheese without coating). CF-40—cheese with commercial formulation coating after 40 days of maturation. M1-40—cheese with melanin film 1 after 40 days of maturation. M2-40—cheese with melanin film 2 after 40 days of maturation.

**Figure 7 foods-12-01806-f007:**
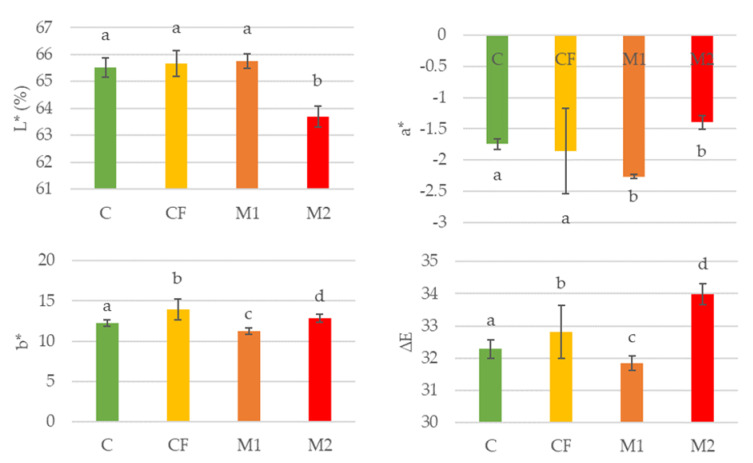
Color parameters and total color difference of cheese rinds with films after 40 days of maturation under industrial conditions. C: cheese control after 40 days of maturation (cheese without coating). CF: cheese with commercial formulation coating after 40 days of maturation. M1: cheese with melanin WPI film 1 after 40 days of maturation. M2: cheese with melanin WPI film 2 after 40 days of maturation. Different lowercase letters (a–d) represent significant differences (*p* ≤ 0.05).

**Figure 8 foods-12-01806-f008:**
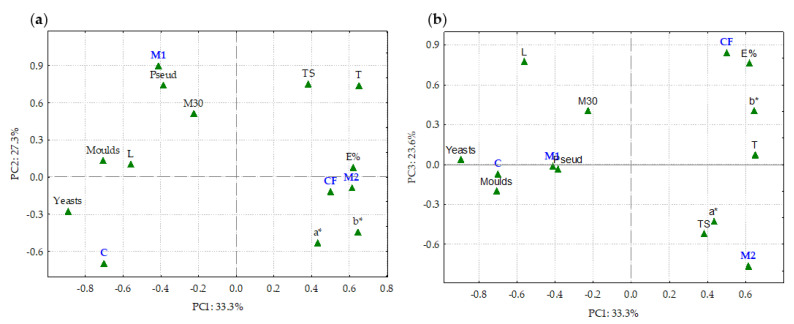
Score plot of the principal components of the PCA (mean-centered) performed with ten variables; (**a**) PC1 vs. PC2 and (**b**) PC1 vs. PC3. C: cheese rind control after 40 days of maturation (cheese without coating). CF: cheese rind with commercial formulation coating after 40 days of maturation. M1: cheese rind with melanin WPI film 1 after 40 days of maturation. M2: cheese rind with melanin WPI film 2 after 40 days of maturation. Microbial parameters: microorganisms at 30 °C (M30), *Pseudomonas* spp. (Pseud), yeasts and molds. Mechanical properties: tensile strength (TS), thickness (T) and extensibility (E%). CIEL*a*b* spaces: lightness (L), red–green (a*) axis and yellow–blue (b*) axis.

**Figure 9 foods-12-01806-f009:**
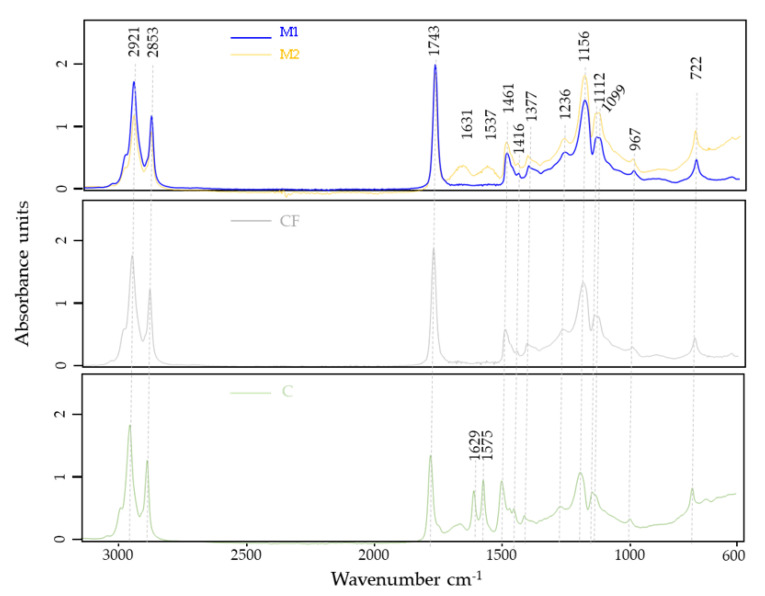
FTIR-ATR spectrum for cheese samples. M1: cheese with melanin WPI film 1 after 40 days of maturation. M2: cheese with melanin WPI film 2 after 40 days of maturation. CF: cheese with commercial formulation coating after 40 days of maturation. C: cheese control after 40 days of maturation (cheese without coating).

**Figure 10 foods-12-01806-f010:**
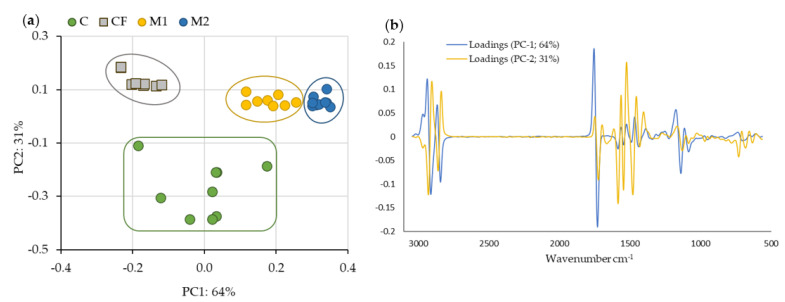
(**a**) Score plot of the first two principal components of the PCA (mean-centered) performed with FTIR-ATR spectra, and (**b**) loading representation of cheese rinds using the first-derivative Savitzky–Golay spectra transform with 17 smoothing points. C: cheese rind control after 40 days of maturation (cheese without coating). CF: cheese rind with commercial formulation coating after 40 days of maturation. M1: cheese rind with melanin WPI film 1 after 40 days of maturation. M2: cheese rind with melanin WPI film 2 after 40 days of maturation.

**Table 1 foods-12-01806-t001:** Assay design of WPI-based studies.

Edible Films	WPI (*w*/*v*)	Plasticizers		Organic Acids	
		Glycerol (*w*/*v*)	Sorbitol (*w*/*v*)	Lactic Acid (*w*/*v*)	Citric Acid (*w*/*v*)
IG3LA	5%	3%	0%	3%	3%
IG3CA	5%	3%	0%	3%	3%
IG5LA	5%	5%	0%	3%	3%
IG5CA	5%	5%	0%	3%	3%
IS3LA	5%	0%	3%	3%	3%
IS3CA	5%	0%	3%	3%	3%
IS5LA	5%	0%	5%	3%	3%
IS5CA	5%	0%	5%	3%	3%

**Table 2 foods-12-01806-t002:** Assay design of melanin WPI coatings under industrial conditions.

	CF (Commercial Film)	M1 (Melanin Film 1)	M2 (Melanin Film 2)
WPI		5%	5%
Sorbitol		3%	3%
Citric Acid		3%	3%
Nisin		50 IU/mL	50 IU/mL
Natamycin		3.5 mg/mL	3.5 mg/mL
Melanin		74 µg/mL	500 µg/mL
VIPLAST-1AX (Concentrol^®^ Chemical Solutions, Girona, Spain)	20%		
RIOCOBERT (BecorBarbanza S.L.U., Coruña, Spain)	30%		

**Table 3 foods-12-01806-t003:** Inhibition zone (mm) of WPI-based films. IG3LA—WPI 5% + glycerol 3% + lactic acid 3%; IG3CA—WPI 5% + glycerol 3% + citric acid 3%; IS3LA—WPI 5% + sorbitol 3% + lactic acid 3%; IS3CA—WPI 5% + sorbitol 3% + citric acid 3%; IS3CAN—WPI 5% + sorbitol 3% + citric acid 3% + natamycin (3.5 mg/mL); IS3CAM—WPI 5% + sorbitol 3% + citric acid 3% + melanin (500 µg/mL); M1—WPI 5% + sorbitol 3% + citric acid 3% + natamycin (3.5 mg/mL) + nisin (50 IU/mL) + melanin (74 µg/mL); M2—WPI 5% + sorbitol 3% + citric acid 3% + natamycin (3.5 mg/mL) + nisin (50 IU/mL) + melanin (500 µg/mL). RIOCOBERT and VIPLAST-1AX are commercial coatings. Different superscript letters A–G correspond to statistically different values (*p* ≤ 0.05) between films (columns). Different superscript letters a–e correspond to statistically different values (*p* ≤ 0.05) between strains inside each film (lines). NI—not inhibited.

Strains	Origin	Inhibition Zone (mm)									
		IG3LA ^A^	IG3CA ^A^	IS3LA ^B^	IS3CA ^C^	IS3CAN ^D^	IS3CAM ^E^	M1 ^F^	M2 ^G^	RIOCOBERT ^D^	VIPLAST-1AX ^D^
*Listeria monocytogenes* NCTC 11944	-	13.0 ^a^	12.8 ^a^	15.1 ^b^	15.7 ^b^	13.0 ^a^	15.4 ^c^	16.8 ^d^	17.2 ^d^	NI	9.2 ^e^
*Staphylococcus aureus* ATCC 25923	-	8.0 ^a^	7.9 ^a^	9.9 ^b^	10.3 ^b^	8.1 ^a^	10.5 ^b^	7.5 ^a^	8.5 ^a^	NI	NI
*Serratia marcescens* ESACB 596	Raw sheep’s milk	9.9 ^a^	9.4 ^a^	11.1 ^b^	10.1 ^b^	8.1 ^a^	11.3 ^b^	10.1 ^b^	10.6 ^b^	NI	NI
*Serratia marcescens* ESACB 734	Raw sheep’s milk	8.8 ^a^	8.4 ^a^	11.8 ^b^	12.8 ^a^	7.9 ^a^	12.9 ^a^	9.8 ^c^	9.1 ^c^	NI	NI
*Pseudomonas putida* ESACB 27	Goat’s cheese	12.9 ^a^	12.7 ^a^	12.9 ^a^	13.6 ^b^	12.5 ^a^	15.8 ^c^	13.9 ^c^	14.5 ^c^	NI	NI
*Pseudomonas fluorescens* ESACB 137	Environment dairy factory	12.8 ^a^	12.8 ^a^	14.3 ^b^	14.6 ^b^	14.8 ^b^	14.9 ^b^	15.6 ^c^	17.4 ^d^	NI	NI
*Pseudomonas putida* ESACB 191	The rind of goat´s cheese	13.6 ^a^	13.3 ^a^	10.7 ^b^	13.9 ^a^	13.8 ^a^	13.9 ^a^	13.4 ^a^	14.4 ^c^	6.3 ^d^	NI
*Pseudomonas fluorescens* ATCC 13525	-	11.8 ^a^	11.8 ^a^	12.8 ^b^	13.5 ^b^	12.4 ^b^	13.9 ^b^	14.1 ^b^	14.5 ^b^	NI	NI
*Pseudomonas aeruginosa* ATCC 27853	-	12.9 ^a^	12.6 ^a^	12.9 ^a^	13.5 ^b^	13.2 ^b^	14.5 ^c^	14.8 ^c^	15.7 ^d^	NI	NI
*Penicillium chrysogenum*	Cheese	NI	NI	NI	NI	28.2 ^a^	NI	31.1 ^b^	32.9 ^b^	27.9 ^a^	31.1 ^b^
*Penicillium commune*	Cheese	NI	NI	NI	NI	27.9 ^a^	NI	25.1 ^a^	32.4 ^b^	27.9 ^a^	24.4 ^a^
*Penicillium roqueforti*	Cheese	NI	NI	NI	NI	30.1 ^a^	NI	25.1 ^b^	28.3 ^a^	24.7 ^b^	23.3 ^b^
Grennish mould	Cheese	NI	NI	NI	NI	27.2 ^a^	NI	23.7 ^b^	24.5 ^b^	18.9 ^c^	16.5 ^c^
White mold	Cheese	NI	NI	NI	NI	22.9 ^a^	NI	17.7 ^b^	19.6 ^a^	17.4 ^b^	13.0 ^b^
*Candida albicans* ATCC 10231	-	NI	NI	NI	NI	NI	NI	20.2 ^a^	20.3 ^a^	15.3 ^c^	13.0 ^c^
*Yarrowia lipolytica*	Cheese	NI	NI	NI	8.70 ^a^	5.5 ^a^	6.7 ^a^	23.1 ^b^	23.9 ^b^	18.5 ^c^	14.4 ^c^
*Candida zeylanoides*	Cheese	NI	NI	NI	NI	NI	NI	7.4 ^a^	7.4 ^a^	NI	17.9 ^b^

## Data Availability

The data presented in this study are available on request from the corresponding author.

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
