# Peer review of "Novel, Edible Melanin-Protein-Based Bioactive Films for Cheeses: Antimicrobial, Mechanical and Chemical Characteristics"

_foods, 2023, doi:10.3390/foods12091806_

Round 1

Reviewer 1 Report

The paper reports the development of a film based on WPI containing melanin which the authors conclude has potential as a suitable edible antimicrobial film for cheese.  It presents some interesting data but its presentation requires considerable improvement.  Many of the problems relate to use of English so a thorough revision by a native English speaker is essential. As indicated below, the meaning of some sentences is unclear, some sentences are out of place and some sentences are irrelevant.

In several places throughout the manuscript, a full stop is used instead of a comma.  These need to be amended.

There is a mix of tenses in the manuscript.  All discussion of results should be in the past tense.

In the Abstract it is stated (lines 32-33) “It was found that the mechanical and colorimetric properties are the ones that most contribute to the distinction of M2 film in cheese from others” Also in the Conclusions, it is stated that “It was found that the mechanical (TS, T, and E%) and colorimetric properties (a* and b*) are the ones that most contribute to the distinction of M2 film in cheese”. I don’t recall seeing data for the mechanical properties of the “M2 film in cheese”. They appear to have been used in the PCA (Figure 8). Please clarify.

I find it curious that one reason why melanin was chosen as an ingredient in the film was because it is “a colorant” (line 140).  In lines 401-403 it is stated that “Color is one of the most important attributes that will be first seen by consumers before eating the cheese and is a primary consideration of consumers when making purchasing decisions”. Also in line 549, the authors consider the M2 film to have “an appealing color for the consumer”. Do the authors seriously consider a cheese with a dark film is going to improve its attraction to consumers? The authors go on to say (lines 403-404) “The color is also an indicator cheese’s quality with attributes such as flavor. sanity. and ripeness”. This sentence is irrelevant given the color of the cheese is not affected by the film’s color.  Further, do the authors really consider that colour of the cheese is related to “sanity”? The sentence should be deleted.

Some specific issues which require the authors’ attention. (line number, comment)

31, I presume “to” should be “the”

48-81, a very long paragraph containing mention of several topics.  Please split into paragraphs with similar content.

115 and elsewhere, delete “organic”; the reader will know citric and lactic are organic acids

130, I believe “better: should be “the best”

132, change “at” to “to”

140, 74,2 seems like an odd number.  Why was this concentration used?

142, I think “released” should be “produced”

143-144, what was thickness of the film? About 1.5 mm?

145, change “film’s” to “films”

179, please explain the significance of the thickness of the film strips

198-199, please indicate what the “standard” was.

203, I believe “diving” should be “dipping”

224-225, please reword the sentence “Both timings were analyzed the same parameters”

234, how were the cheese samples “mixed”?

253, what is meant by “did not an interaction ration”

252-262.  this paragraph needs reorganising.  The sentence in lines 261-262 should be at the beginning. The first two sentences of the current paragraph could be deleted as the information has already been covered.

258, Figure 3 does not indicate this

264-267, please correct all film abbreviations in this caption. For example, IG3AC should be IG3CA

270, “de” ??

274, please explain how “sorbitol increases the film thickness”

278-281, seems to be out of place here.  What blocking agent? Is this related to extensibility? What is the evidence for the statement. Consider deleting or moving these sentences.

282-284, please reword. LA and CA do not decrease or increase extensibiity; suggest “CA films were more extensible than LA films”

284-286, because CA is a larger molecule doesn't mean it lowers the pH more than LA

296-298, please explain how this relates to this paper.

301, what is meant by “All films are transparent color, so, it is normal higher L* values.”

305, what is meant by “adding a better visual appearance”. Is it being transparent?

312, suggest change “glycerol films and sorbitol” to “glycerol and sorbitol films”

314-316, clumsy wording. Suggest change to “IS3CA films had significantly higher antimicrobial activity than IS3LA films”.

319-320, this sentence needs more information to be informative to the reader.

322, delete “than the other films”. Repeated

325, please supply reference for “the reduced ability of citric acid to enter bacterial cells”

331, suggest change “activity that already” to “agent that is already”

339-340, Is this an average of all strains? Is it for the high or low melanin level? Please clarify

341-342, this sentence does not make sense, please reword

344-346, this sentence does not make sense, please reword

347, incorrect reference number

350, I am confused as to whether one or two commercial films were used. In lines 375-376, the term “the commercial formulation” is used, suggestion only one film. Please clarify

350-351, poor English. Please reword

Table 2, to make this table more readable, I suggest citing means only, i.e., remove SDs, and decrease the number of decimal places to one

359, I believe the films were applied after production not during

377, suggest replace “With the maturation fulfilled” by “after maturation”

377-381, this information is repeated from the Material and Methods section

397-398, “as happened in antimicrobial test (Table 2)” can be replaced with “as shown in Table 2”

408, ΔE may be “a good measure” but is a high value good or not?

418-420, I suggest deleting these sentences. There is no relationship between absolute values of L and a and b, so these sentences make no sense

424-426, this is out of place here

427-431, this information is repeated from lines 420-423. Please delete.

Section 3.4, I see little value in this section. 

512-513, not a sentence

514-515, I cannot see how this conclusion can be reached from the FTIR data.

534, I suggest adding “treated cheese” after “M2”

Reviewer 2 Report

It seems to be a novel idea, but the experimental design and presentation are very low-level, with a large amount of necessary information being non-standard or missing, making it difficult for readers to find more valuable information from the manuscript. Authors need help from experienced experimenters and language users, to systematically improve this manuscript. At least major revisions are required.

1. L2, L26: the full name of WPI should be given at its first mention.

2. L25-26: as generally known, the quality of the cheese cannot only be represented by color defects.

3. L40: living?

4. L56-58: but antimicrobial content used in this work, including sorbitol and citric acid can be considered as preservatives.

5. Introduction: Logic is confusing and a large number of descriptions seem not very related to this work. Please rewrite. Your focus should be put on the current research status and bottlenecks, as well as the advantages of your research.

6. The design of Figure 1 should be improved. Maybe you can use a table.

7. L114, 121: it is not necessary to give a full name if it is not mentioned for the first time.

8. L141: more detail about melanin should be given. Is it from your laboratory? If so,  how did you obtain it from Pseudomonas putida ESACB 191?Or from reagent merchant? Thus, please give company, model, purity, etc.

9. L142: filmogenic solution, what is it?

10. L147: of which 12 belong to the microbial culture collection..., what are them?

11. L156: Table 1 is missing.

12. L182-183: “A TA.XT plus test machine (Stable Micro Systems) equipped with traction grippers (model A/TG)”. Standard format to introduce an instrument is given in the Guide for Authors.

13. L191: the CIELAB color system

14. The ratio of each component in the films seems to be set too arbitrarily and for no reason

15. Materials and Methods: A large amount of content is non-standard or difficult to understand. Some necessary details are missing. A significant improvement is required.

16. Results and Discussion: The data presentation seems incomplete. Some necessary data are missing. Many trends cannot summarize an applicable rule and lack relevant mechanism analysis. Actually, the properties differences between different samples seem to be insignificant.

Round 2

Reviewer 1 Report

Most of the comments in my original review have been satisfactorily addressed.  However, the following points require further attention by the authors.

In answer to the query regarding the data for mechanical strength of the M1 and M2 films, the authors have now included the relevant data in Table 3 not Table 4 as stated in their cover letter. 

The response to the query “I am confused as to whether one or two commercial films were used. In lines 375-376, the term “the commercial formulation” is used, suggestion only one film. Please clarify” contains important information about the commercial films which needs to be included in the manuscript.

The authors’ response to query “377-381, this information is repeated from the Material and Methods section” was “The sentence was reworded to “The analytic results are of microbial growth on the rind of the cheeses during maturation are shown in Figure 6””. This statement is untrue as the change has not been made to the manuscript.

In response to query “514-515, I cannot see how this conclusion can be reached from the FTIR data.” is unsatisfactory.  I suggest the sentence, now in lines 617-618, be deleted as FTIR cannot show that melanin is important for control of the quality of cheese.